# Depression and Suicidal Ideation in Patients with Mental Illness in South Korea: The Mediating Effect of Alcohol Drinking

**DOI:** 10.3390/healthcare11202711

**Published:** 2023-10-11

**Authors:** Kyoungsook Lee, Oisun Lee

**Affiliations:** 1Department of Nursing, University of Ulsan, Ulsan 44610, Republic of Korea; gslee1@ulsan.ac.kr; 2Department of Nursing, Gyeongnam Geochang University, Geochang 50147, Republic of Korea

**Keywords:** depression, alcohol drinking, suicide, survey

## Abstract

This study aimed to investigate the mediating effect of alcohol drinking on the relationship between depression and suicidal ideation inpatients with mental illness. A survey was conducted among 163 individuals with various major mental disorders using a self-reported questionnaire. Data were collected from July to September 2020. Subsequently, the data were analyzed using a t-test, one-way analysis of variance, Pearson’s correlation coefficients, hierarchical multiple linear regression, and a Sobel test. Significant relationships were found between alcohol depression and drinking (r = 0.26, *p* < 0.001), depression and suicidal ideation (r = 0.63, *p* < 0.001), and alcohol use and suicidal ideation (r = 0.36, *p* < 0.001). In addition, alcohol drinking was found to partially mediate the association between depression and suicidal ideation (Z = 3.63, *p* < 0.001). These results indicated that patients with mental illness who are concerned about drinking alcohol might be able to reduce suicidal thoughts by consulting with a healthcare professional or taking informed actions with the support of a counselor or support group.

## 1. Introduction

In South Korea, the suicide rate was 26 per 100,000 people in 2021, 1.1% higher than that reported in 2020 [1]. Approximately 90% of the suicide deaths were associated with mental illness; psychiatric patients were found to have a noticeably higher suicidal rate than normal individuals [2]. According to domestic and foreign reports, mental disorders are the main cause of suicide [3,4]. Therefore, it is possible to understand that mental health and suicide are closely related [3,4]. Mental health disorders are a pervasive global health concern, with depression being one of the leading causes of disability worldwide [5]. Among individuals dealing with psychiatric conditions, the co-occurrence of depression and suicidal ideation presents a particularly alarming challenge for both clinicians and patients. While several studies have explored the intricate relationship between depression and suicidal ideation [5], a lesser-known factor that may contribute to this complex dynamic is alcohol drinking.

Suicide attempts can often lead to death. Therefore, preventing suicidal behavior is important. In general, suicide begins with suicidal ideation, progresses to suicidal attempts, and ends up completing the suicide [6].Thus, understanding suicidal ideation at an early stage is important before it progresses to suicidal behavior and leads to suicidal attempts [6]. In particular, regarding the relationships between suicide and representative chronic mental disorders (schizophrenia, depression, and bipolar disorder), approximately 50% of patients with schizophrenia attempted suicide, of which 10% died [7,8]. Moreover, approximately 2–15% of patients with major depression, 3–20% of those with bipolar disorder, and 6–15% of those with schizophrenia have been reported to commit suicide [2]. In addition, depression is a pervasive and debilitating mental health condition that impacts millions of individuals globally [9]; therefore, studying the relationship between depression and suicidal ideation is necessary. Suicidal ideation is associated with depression, and depression severity affects suicidal ideation [9,10]. This is because depression impairs problem-solving skills; thus, patients with depression choose suicide as a maladaptive attempt to be free from their depressive emotions and problems [11]. The increasing prevalence of depression in South Korea is a concern, imposing a significant burden on such patients. Moreover, untreated depression can lead to severe consequences, including suicidal ideation [11].

Alcohol drinking is another suicidal factor. It is persistently involved in suicidal behavior [12]. Moreover, suicidal behavior has been reported to likely occur in patients having both depression and the habit of alcohol drinking [13]. This is because alcohol drinking damages one’s judgment and hope and thus increases suicide risk [14]. Once a suicidal attempt is determined, alcohol drinking plays a critical role in behavior [15]. Hence, the suicidal attempt can be used as “a means to an end”. Moreover, the mutual relationship between alcohol drinking and suicidal behavior is possible through other variables rather than having a causal relationship. In other words, regarding the relationship between alcohol drinking and suicidal behavior, the premise is that individuals’ internalized or externalized psychopathology frequently affects alcohol drinking and suicidal behavior and can influence the relationship [16,17].

In the case of patients with mental illness, depression is closely related to suicidal ideation and alcohol drinking. Accordingly, alcohol drinking can have mediating effects on the relationship between depression and suicide. In this aspect, conducting a direct empirical study on the effect of alcohol drinking on the association between depression and suicide is necessary. Therefore, this study attempted to investigate depression levels, suicidal ideation, and alcohol drinking among patients with mental illness. Moreover, the effects of alcohol drinking on suicidal ideation and the mediating effect of alcohol drinking on the relationship between these factors were also investigated. The findings of this study provide basic data to establish a strategy to reduce suicidal ideation in patients with mental illness.

Understanding the role of alcohol drinking in patients with mental illness is essential for designing targeted interventions. By identifying alcohol drinking as a potential mediator, this study may aid in the development of more effective treatment strategies that consider cultural and social factors specific to South Korea. It can also contribute to the broader global understanding of how alcohol influences the complex relationship between depression and suicidal ideation.

Depression and suicidal ideation are pressing concerns in South Korea, and the role of alcohol drinking in mediating these concerns needs to be thoroughly examined. This study seeks to shed light on the relationship between depression and suicide, thereby providing valuable insights for mental health professionals, policymakers, and researchers striving to improve the well-being of patients with mental illness in South Korea and worldwide.

This study aimed to verify the levels of depression, alcohol drinking, and suicidal ideation among patients with mental illness, the correlations between the variables, and the factors influencing suicidal ideation. In particular, it explored whether alcohol drinking mediates the relationship between depression and suicidal ideation in this unique cultural setting.

## 2. Methods

### 2.1. Study Design

This study is a descriptive survey research to verify the degree of depression, alcohol drinking, and suicidal ideation in psychiatric patients, the correlations among these variables, and factors affecting suicidal ideation (Figure 1).

### 2.2. Study Population

Regarding the number of study subjects, the G-power 3.1.9 program was applied to calculate an appropriate number of samples. The minimum number of samples for multiple regression analysis was 147 when the significance level was 0.05, explanatory power was 0.90, medium effect size was 0.15, and the number of predictive factors was 12.

From July to September 2020, a total of 280 individuals from five basic mental health welfare centers, two addiction management support centers, two rehabilitation facilities, and eight mental health medical centers were enrolled through convenient sampling. Prior to conducting the questionnaire survey, an official letter of cooperation was sent, and approval was received from the institutions. Among the individuals who volunteered to participate in the study, the ones suitable for participation in the survey were chosen. In particular, 141 members of mental health improvement centers and 99 inpatients from mental health medical centers responded to the questions in the questionnaire survey. Of them, those who chose multiple answers and did not respond honestly were excluded from the study. Of the 212 respondents, 30 with alcohol use disorders and 9 with anxiety disorders were excluded. Finally, data from 163 patients were analyzed. The inclusion criteria of the participants were as follows: (1) those aged 18–65 years; (2) those diagnosed with schizophrenia, depression, and bipolar disorder by a psychiatrist according to the diagnostic criteria of the Diagnosis and Statistics Manual (DSM)-5 in American Psychiatric Association; (3) those residing in South Korea or receiving psychiatric treatment in South Korea, Ulsan metropolitan city; (4) those with no acute symptoms (hallucinations and delusions) or behavioral disorders (self-harm, other harm disorder); (5) those without mental retardation or psychiatric double diagnosis; (6) those who are medically free of organic brain damage or nervous system disease; (7) those who can perform their daily activities easily; (8) those who provided informed consent to participate in the study; (9) those who were able to understand and communicate in Korean language. The exclusion criteria of the participants were as follows: (1) those who did not provide informed consent to participate in the study; (2) those who were deemed incapable of understanding the study or providing valid responses due to severe cognitive impairment or intoxication at the time of assessment; (3)those with a history of alcohol use disorders, drug use disorders, and multi matter disorders or who were recently diagnosed with any of these disorders; (4) those with medical conditions that may significantly affect their psychiatric symptoms or confound the study’s results (e.g., those with severe traumatic brain injury); (5) those who could not communicate effectively in Korean language; (6) those below 18 years of age or over 65 years of age; (7) those who were pregnant or consumed alcohol (drugs); (8) those who faced difficulties in performing daily activities.

### 2.3. Measures

#### 2.3.1. Variables

This study used a structured self-report questionnaire that comprised 45 questions, including 10 questions on general characteristics, 20 on depression, 10 on alcohol use, and 5 on suicidal ideation.

#### 2.3.2. Depression

To measure the level of depression inpatients with mental illness, the depression scale for the Center for Epidemiological Studies (Center for Epidemiological Studies Depression Scale) was used [18], which was developed by the US National Institute of Mental Health and was translated by Cho and Kim [19]. The 4-point Likert scale consists of 20 items about a respondent’s feelings over the past week. The scores on the scale range from 0 to 60. The higher the score, the higher the level of depression. In a study by Cho and Kim [19], Cronbach’s α was 91; however, in the present study, Cronbach’s α was 0.94.

#### 2.3.3. Alcohol Drinking

To measure the level of alcohol drinking inpatients with mental illness, the Alcohol Use Disorders Identification Test was used [20], which was developed by the WHO and translated by Lee et al. [21]. The test consists of 10 items about a respondent’s alcohol use over the past year. The higher the score, the greater the problem of alcohol drinking. In this study, Cronbach’s α was 0.95.

#### 2.3.4. Suicidal Ideation

To evaluate suicidal ideation inpatients with mental illness, the Suicidal Ideation Scale was used, which was developed by Harlow et al. [22] and translated and modified by Kim [23]. The scale consists of five items about are spondent’s suicidal ideation over the past year. The higher the score, the greater the suicidal ideation. In this study, Cronbach’s α was 0.94.

### 2.4. Data Collection

After receiving approval from the Institutional Review Board of the University of Ulsan (IRB NO: 2020R0012) in July 2020, a questionnaire survey was conducted for approximately 3 months from July to September 2020. According to a previous study, if a questionnaire survey is conducted without the formation of patients’ trust with mental illness, obtaining honest answers is difficult due to their uneasiness; therefore, allowing the staff of related institutions to perform the survey directly is necessary [24,25]. In this study, the staff of the participating centers conducted the questionnaire survey directly. The staff were taught about the roles of survey agents, including the criteria of subject selection and attention points for a questionnaire survey, and offered return gifts by the researcher. If study participants decided to join the survey and met the selection criteria, each one of the survey agents who had knowledge about the survey explained the study procedure and content to them. After having agreed to the contents of the study, a questionnaire survey was conducted on the study participants; once the survey was completed, the researcher provided return presents and brochures describing how to respond to negative psychological reactions that could arise after the survey. The researcher visited each center and collected the questionnaire data directly.

### 2.5. Data Analysis

The data collected in this study were analyzed using the SPSS/WIN 27.0 program. The statistically significant level was 0.05. The values of frequency, percentage, mean, and standard deviation were calculated to measure the subjects’ general characteristics, depression, alcohol drinking, and suicidal ideation. To identify the differences in general characteristics, a t-test and one-way analysis of variance were performed, and the Scheffé test was conducted as a post hoc test. The correlations between the subjects’ depression, alcohol drinking, and suicidal ideation were analyzed using Pearson’s correlation coefficients. Hierarchical regression analysis was performed to analyze the factors influencing the subjects’ suicidal ideation. The mediating effect of alcohol drinking on the relationship between the subjects’ depression and suicidal ideation was analyzed using the multiple regression analysis-based three-step mediation analysis approach of Baron and Kenny [26]. The Sobel test was conducted to determine the statistical significance of the mediating effect.

## 3. Results

### 3.1. General Characteristics of the Participants and the Difference of Suicidal Ideation Depending on their General Characteristics

The general characteristics of the study participants were analyzed. Men accounted for 50.3% of the study participants. The mean age of the participants was 44.10 ± 13.29 years; those in their 40s accounted for the highest percentage (24.8%), and those in their 20s to 60s were distributed evenly. Regarding religion, 50.3% of the study participants replied that they had no religion. Regarding hospitalization status, 58.4% stayed in local communities. Regarding DSM-5-based diagnosis, 50.3% had schizophrenia, 30.1% had depression, and 19.6% had bipolar disorder. Regarding the first onset, 30.3% had the first onset at the age of <20 years, and 27.6% had it at the age of ≥40 years. Regarding the number of admissions, 31.4% experienced no admission. Regarding daily average sleep time, 31.9% had 7–8 h of sleep, 30.9% had over 9 h of sleep, and 21.1% had less than 4 h of sleep. Regarding sleep satisfaction, 34.1% were neither satisfied nor dissatisfied, 33.5% were not satisfied, and 32.4% were satisfied. Regarding previous suicide attempts, 50.3% had not attempted suicide (Table 1).

Regarding the differences in suicidal ideation depending on general characteristics, significant differences were found in gender (t = −2.84, *p* = 0.005), diagnosis (F = 22.55, *p* < 0.001), sleep time (F = 7.06, *p* < 0.001), sleep satisfaction (F = 16.44, *p* < 0.001), and previous suicide attempts (F = 6.64, *p* < 0.001) (Table 1). With regard to the difference in suicidal ideation by gender, women had more suicidal ideation than men; with regard to the difference in suicidal ideation by diagnosis, those diagnosed with depression had more suicidal ideation than those with schizophrenia or bipolar disorder; with regard to the difference in suicidal ideation by sleep time, those who had less than 4 h of sleep per day had more suicidal ideation than those who had 5–6 h of sleep, 7–8 h of sleep, and over 9 h of sleep; regarding sleep satisfaction, those dissatisfied with sleep had more suicidal ideation than those satisfied with sleep and those neither satisfied nor dissatisfied with sleep; and regarding previous suicide attempts, those who experienced any previous suicide attempts had more suicidal ideation than those with no experience of previous suicide attempts. No significant difference was found in suicidal ideation based on age, religion, the first onset, and the number of admissions.

### 3.2. Degrees of Depression, Alcohol Use, and Suicidal Ideation

Depression in the study participants was measured. Out of 60, the mean ± SD score was 26.97 ± 13.62. Out of 40, alcohol drinking scored 4.32 ± 8.48. Out of 20, suicidal ideation scored 9.15 ± 4.69 (Table 2).

### 3.3. Correlations between the Study Participants’ Depression, Alcohol Use, and Suicidal Ideation

The study participants’ depression was significantly positively correlated with alcohol drinking (r = 0.26, *p* < 0.001) and suicidal ideation (r = 0.63, *p* < 0.001). Moreover, alcohol drinking was significantly positively correlated with suicidal ideation (r = 0.36, *p* < 0.001) (Table 3).

### 3.4. Effects of the Study Participants’ Depression and Alcohol Drinking on Suicidal Ideation

The factors influencing the subjects’ suicidal ideation were analyzed using the hierarchical regression analysis approach. The results are presented in Table 4. In Step 1, general characteristics were used as control variables. In Step 2, depression was applied. In Step 3, depression and alcohol drinking were applied. General characteristics, including gender, age, religion, hospitalization status, diagnosis, first onset, the number of admissions, sleep time, sleep satisfaction, and previous suicide attempts, were processed as dummies. In Step 1 where control variables were applied, gender (β = 0.16, *p* = 0.007), age (β = −0.18, *p* = 0.003), diagnosis (β = 0.25, *p* < 0.001), sleep time (β = −0.14, *p* = 0.030), sleep satisfaction (β = 0.14, *p* = 0.043), and previous suicide attempts (β = −0.36, *p* < 0.001) significantly influenced suicidal ideation. The explanatory power in Step 1 was 42.6%. In Step 2, control variables were controlled, and depression was applied. Of the variables used in Step 1, gender (β = 0.11, *p* = 0.044), age (β = −0.17, *p* = 0.002), diagnosis (β = 0.18, *p* = 0.004), previous suicide attempts (β = −0.28, *p* < 0.001), and depression (β = 0.43, *p* < 0.001) significantly affected suicide. The explanatory power in Step 2 was 52.3%, 9.7% higher than that in Step 1. In Step 3, control variables were controlled, and depression and alcohol drinking were applied. Accordingly, gender (β = 0.14, *p* = 0.009), age (β = −0.19, *p* = 0.002), diagnosis (β = 0.13, *p* = 0.031), previous suicide attempts (β = −0.26, *p* < 0.001), depression (β = 0.42, *p* < 0.001), and alcohol drinking (β = 0.17, *p* = 0.005) significantly influenced suicide. The explanatory power in Step 3 increased by 2% from that in Step 2. Therefore, regarding the influence of depression and alcohol drinking on suicidal ideation, the explanatory power was 54.3%. Furthermore, the presence of multicollinearity between variables was analyzed. As a result, the variance inflation factor value was 1.01–1.45. No multicollinearity was found between variables lower than 10. Hierarchical regression analysis showed that the value of Durbin–Watson was 1.811. Therefore, the correlation between errors had no problem.

### 3.5. The Mediating Effect of Alcohol Drinking on the Relationship between the Study Participants’ Depression and Suicidal Ideation

Regression analysis was performed on the mediating effects in this study. The results are presented in Table 5. The mediating effect of alcohol drinking on the relationship between depression and suicide in patients with mental illness was analyzed. Hence, in Step 1, the independent variable “depression” statistically and significantly affected the mediating variable “alcohol drinking” (β = 0.26, *p* = 0.001). In Step 2, the independent variable “depression” statistically and significantly influenced the dependent variable “suicide” (β = 0.63, *p* < 0.001). In Step 3, depression and the mediating variable “alcohol drinking” as independent variables and “suicide” as a dependent variable were used. As a result, depression and alcohol drinking significantly influenced suicide (β = 0.57, *p* < 0.001). A comparison of regression coefficients demonstrated that the influential power of the independent variable “depression” on suicide in Step 2 (β = 0.63) was lower than that in Step 3 (β = 0.57). Therefore, alcohol drinking partially mediated the relationship between depression and suicide. The value of R^2^ that represents explanatory power was 6.3%, 39.8%, and 43.5% in steps 1, 2, and 3, respectively. The Sobel test was performed to verify the significance of the mediating effect coefficients. As a result, the absolute value of Z was larger than 1.96; thus, the mediating effect of alcohol drinking was found to be statistically significant (Z = 3.63, *p* < 0.001).

## 4. Discussion

Depression and suicidal ideation are significant mental health concerns worldwide, including in South Korea. This study aimed to explore the relationship between depression, suicidal ideation, and alcohol drinking in patients with mental illness in South Korea, with a focus on the mediating effect of alcohol drinking. Understanding these complex interactions is crucial for developing effective interventions and support systems for individuals facing these challenges.

In this study, the factors influencing suicidal ideation in patients with mental illness included depression, previous suicide attempts, age, alcohol drinking, gender, and diagnosis. Prior to analyzing the mediating effect of alcohol drinking, the effects of depression and alcohol drinking on suicidal ideation were analyzed. Consequently, depression and alcohol drinking were found to significantly affect suicidal ideation; this finding was consistent with the results of Tidemalm [27], Kim et al. [9], and Lee et al. [10], who showed that depression in psychiatric patients affected their suicide. Therefore, depression was considered to be a risk factor for suicide. This study demonstrated that alcohol drinking significantly affected suicidal ideation; this finding was consistent with that of Heo et al. [28], who revealed that alcohol use in patients with mental illness increased their suicidal ideation. In addition, alcohol drinking has been reported to be the main predictive variable of suicide [29,30], as revealed in this study. Depression is a mental disorder in which the activities of daily living are hampered and is the most common condition for patients with mental illness [27]. The participants of this study have had regular psychotherapy as outpatients and have received case care from mental health promotion centers for depression. Nevertheless, the average depression score of the subjects in this study was 26.97, indicating that they were suffering from severe depression. Depression was found to be more severe in patients with depression than in those with schizophrenia or bipolar disorder; therefore, more attention needs to be paid to reducing suicidal thoughts in subjects diagnosed with depression. In addition, no difference in the degree of depression was observed between inpatients and those living in the community, so suicide attempts should be prevented by paying more attention to the suicidal thoughts of all those diagnosed with depression.

This study showed that alcohol drinking partially mediated the association between depression and suicidal ideation, indicating that the higher the depression is, the greater the level of suicidal ideation is, and the higher the use of alcohol is, the greater the level of suicidal ideation is. In other words, excessive alcohol drinking increases the risk of suicide by causing psychological devastation, social isolation, economic loss, and family troubles [31]. Moreover, alcohol drinking makes it difficult to control emotions and make rational decisions, thereby reducing one’s internal ability to overcome negative feelings and prolonging negative thoughts and pessimistic views. Furthermore, it enhances suicidal plans or suicidal impulses, increasing the likelihood of attempting impulsive suicide [32]. Yoon et al. [33] reported that the behavior of alcohol drinking had a negative effect on suicidal ideation longitudinally. The COVID-19 pandemic and its associated stressors might have resulted in the use of alcohol by individuals, including psychiatric patients, as a coping mechanism. Alcohol can provide temporary relief from emotional distress, but it is not a long-term solution and can exacerbate mental health issues. Due to the restriction of social activities during the pandemic, individuals may have had more idle time at home, leading to an increase in the opportunity for alcohol consumption as a form of entertainment or relaxation. Given that long-term consumption of alcohol can affect suicidal ideation, providing support and education is necessary for psychiatric patients to give up drinking.

Depression in patients with mental illness was found to indirectly affect suicidal ideation by way of increasing alcohol drinking. In other words, alcohol drinking increases with depression, which causes impulsive suicidal behavior through suicidal ideation. Previous studies [34,35] have reported that alcoholics are more likely to commit suicide. Given all these findings, it is necessary to educate psychiatric patients on giving up drinking alcohol to reduce suicidal ideation. In particular, since patients with mental illness continue to use alcohol and have less motivation to reduce alcohol use [36], selecting an alcohol-drinking group is necessary for making early interventions and developing and providing an efficient alcohol-use-reduction program. Gregg et al. [37] reported that suicidal ideation is related to alcohol use in patients with mental illness. Depression and suicidal ideation are serious mental health problems in Korea, and alcohol drinking has been found to partially mediate suicide. Understanding the mediating effects of alcohol drinking in this context is essential for effective intervention and support. Combining culturally sensitive approaches with evidence-based strategies can help solve these complex problems and improve the mental health of Korean patients with mental illness.

The present study found compelling evidence to support the hypothesis that alcohol drinking mediates the relationship between depression and suicidal ideation among patients with mental illness. This suggests that alcohol drinking serves as a mechanism through which depressive symptoms contribute to the emergence or exacerbation of suicidal thoughts in this vulnerable population. The findings underscore the importance of holistic mental health care that considers the interplay of multiple factors in patients with mental illness. This approach should encompass not only the treatment of depressive symptoms but also the identification and management of alcohol use disorders.

Further research is needed to understand in depth the dynamics of depression, suicidal thoughts, and drinking in Korea. In addition, to reduce suicide accidents, it is necessary to develop targeted prevention and treatment programs to reduce depression and alcohol consumption. Moreover, the COVID-19 pandemic might have exacerbated the challenges faced by patients with mental illness in South Korea, including depression and suicidal ideation. The mediating effect of alcohol drinking, as a potential coping mechanism during the pandemic, should be further studied to develop effective interventions and support systems for individuals dealing with mental health issues in such challenging times. Providing accessible mental health services and addressing substance use issues are essential components of this effort.

## 5. Conclusions and Limitations

The limitation of this study is the possibility of generalization. Data collection and analysis were performed targeting only one city; hence, interpreting and generalizing these data to every patient with mental illness needs to be performed with caution.

This study verified that among psychiatric patients, alcohol drinking partially mediated the association between depression and suicidal ideation. Therefore, to prevent suicide among patients with mental illness, active intervention is necessary to help them reduce their depression and give up drinking.

While this study provides valuable insights, it is essential to recognize that the relationship among depression, suicidal ideation, and alcohol drinking is complex. Further research is warranted to explore the nuances of this relationship, including the impact of specific types of alcohol and the effectiveness of tailored interventions.

In this study, the association between depression, alcohol drinking, and suicide and the mediating effect of alcohol drinking was confirmed. However, it was not possible to confirm the reverse association between depression and alcohol. Therefore, in future studies, it will be suggested to study the reverse association between depression and alcohol.

In conclusion, this study highlights the role of alcohol drinking as a mediating factor between depression and suicidal ideation among patients with mental illness. It emphasizes the importance of personalized, comprehensive mental health care that addresses both depressive symptoms and alcohol misuse. The findings of this study have practical implications for risk assessment, intervention strategies, and the overall well-being of individuals grappling with these challenging conditions. While this study focuses on South Korea, the findings can also be used to understand the relationship between depression, suicidal ideation, and alcohol use in patients in other countries, especially those with similar societal and cultural characteristics.

## Figures and Tables

**Figure 1 healthcare-11-02711-f001:**
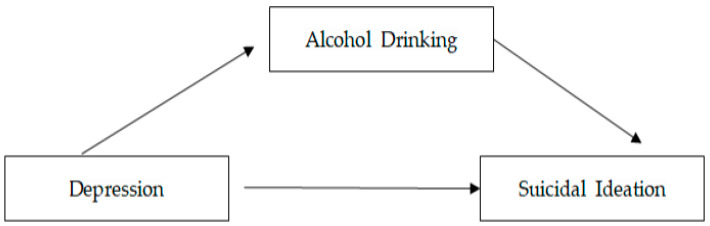
Conceptual framework.

**Table 1 healthcare-11-02711-t001:** Subject characteristics and differences in suicidal ideation by general characteristics (*n* = 163).

Characteristics	Category	*n*	%	Mean ± SD	Suicidal Ideation	
					Mean ± SD	t/F (*p*) Scheffé Test
Gender	Male	81	49.7		8.08 ± 3.92	−2.84 (0.005)
	Female	82	50.3		10.09 ± 5.11	
Age (years)	20 s	35	21.5	44.10 ± 13.29	10.86 ± 5.21	1.71 (0.149)
	30 s	25	15.3		8.13 ± 4.10	
	40 s	42	24.8		9.61 ± 5.18	
	50 s	35	21.5		8.64 ± 4.27	
	≥60 s	26	16.9		8.25 ± 3.84	
Religion	Have	82	50.3		8.84 ± 5.07	0.95 (0.341)
	None	81	49.7		9.57 ± 5.261	
Hospitalization status	Inpatient	68	41.6		8.60 ± 4.57	0.16 (0.850]
	Outpatient	95	58.4		9.44 ± 4.75	
Diagnosis	Schizophrenia ^a^	82	50.3		7.34 ± 3.59 ^a^	22.55 (<0.001) c > a,b
	Bipolar disorder ^b^	32	19.6		8.84 ± 4.59 ^a^	
	Depression ^c^	49	30.1		12.39 ± 4.75 ^c^	
First onset	<20	49	30.3	30.35 ± 13.71	7.91 ± 3.86	2.18 (0.091)
	20 s	33	20.0		10.06 ± 5.54	
	30 s	36	22.1		9.89 ± 4.25	
	≥40	45	27.6		9.67 ± 4.91	
Number of admissions	0	51	31.4	2.91 ± 1.54	7.91 ± 3.86	1.85 (0.123)
	1	31	18.9		10.06 ± 5.54	
	2	36	22.1		9.89 ± 4.25	
	≥3	45	27.6		9.67 ± 4.91	
Sleep time	≤4 ^a^	34	21.1	7.74 ± 2.33	12.29 ± 5.67	7.06 (<0.001) a,b > c,d
	5–6 ^b^	26	16.2		10.37 ± 5.32	
	7–8 ^c^	52	31.9		7.90 ± 3.52	
	≥9 ^d^	51	30.8		8.51 ± 4.33	
Sleep satisfaction	Satisfaction ^a^	53	32.4		7.40 ± 3.62 ^a^	16.44 (<0.001) c > a,b
	Neutral ^b^	56	34.1		8.78 ± 4.36 ^b^	
	Dissatisfied ^c^	54	33.5		12.37 ± 5.06 ^c^	
Previous suicide attempts	Have	82	50.3		11.30 ± 4.96	6.64 (<0.001)
	None	81	49.7		6.98 ± 3.22	

**Table 2 healthcare-11-02711-t002:** Degree of variables (*n*= 163).

Variable	Mean ± SD	Actual Range	Reference Range
**Depression**	26.97 ± 13.62	0–58	0–60
Slowdown in physical behavior	11.20 ± 6.15	0–24	0–24
Depressive emotion	9.55 ± 5.61	0–21	0–21
Positive emotion	4.10 ± 2.05	0–9	0–9
Interpersonal relationship	2.12 ± 1.79	0–6	0–6
**Alcohol** **drinking**	4.32 ± 8.48	0–40	0–40
**Suicide ideation**	9.15 ± 4.69	5–20	5–20

**Table 3 healthcare-11-02711-t003:** Correlation among depression, alcohol drinking, and suicidal ideation (*n* = 163).

Variable	Depression	Alcohol Drinking	Suicide Ideation
	r(*p*)	r(*p*)	r(*p*)
Depression	1		
Alcohol drinking	0.26 (<0.001)	1	
Suicide ideation	0.63 (<0.001)	0.36 (<0.001)	1

**Table 4 healthcare-11-02711-t004:** Effects of depression and drinking on suicidal ideation (*n* = 163).

Variable	Model 1			Model 2			Model 3		
	B	β	*p*	B	β	*p*	B	β	*p*
Gender	1.57	0.16	0.007	1.07	0.11	0.044	1.38	0.14	0.009
Age	−0.69	−0.18	0.003	−0.639	−0.17	0.002	−0.63	−19	0.002
Religion	0.03	0.14	0.623	0.03	0.03	0.580	0.02	0.02	0.659
Hospitalization status	0.02	0.11	0.666	−0.01	−0.01	0.846	−0.01	−0.01	0.937
Diagnosis	1.3	0.25	<0.001	0.968	0.18	0.004	0.73	0.13	0.031
First onset	0.03	0.03	0.620	0.04	0.04	0.432	0.04	0.05	0.393
Number of admissions	−0.02	−0.03	0.603	−0.02	−0.02	0.611	−0.03	−0.03	0.558
Sleep time	−0.64	−0.14	0.030	−0.10	−0.10	0.090	−0.08	−0.09	0.127
Sleep satisfaction	0.85	0.14	0.043	0.07	0.08	0.163	0.07	0.07	0.250
Previous suicide attempt	−3.4	−0.36	<0.001	−2.62	−0.28	<0.001	−2.43	−0.26	<0.001
Depression				0.15	0.43	<0.001	0.14	0.42	<0.001
Alcohol use							0.09	0.17	0.005
R^2^	0.447			0.537			0.560		
Adj. R^2^	0.426			0.523			0.543		
F change	21.03			36.46			33.08		
Durbin–Watson							1.81		

**Table 5 healthcare-11-02711-t005:** The mediating effect of alcohol drinking on the relationship between depression and suicidal ideation (*n* = 163).

Step	Independent Variables	Dependent Variables	B	S.E	β	T(*p*)	R^2^	Adjusted R^2^	F (*p*)
1	Depression	Alcohol drinking	0.16	0.04	0.26	3.46(<0.001)	0.069	0.063	11.98 (<0.001)
2	Depression	Suicidal ideation	0.21	0.02	0.63	10.38(<0.001)	0.401	0.398	107.92 (<0.001)
3	Depression, alcohol drinking	Suicidal ideation	0.19	0.02	0.57	9.45(<0.001)	0.442	0.435	63.38 (<0.001)
			0.12	0.034	0.21	3.42(<0.001)			
Sobel test Z(*p*) = 3.63 (*p* < 0.001)									

## Data Availability

The data sets used and analyzed in the current study are available from the first author on reasonable request. We confirm that this is the case, and ethical considerations or privacy regulations prevent us from sharing the data.

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
