# Peer review of "Depression and Suicidal Ideation in Patients with Mental Illness in South Korea: The Mediating Effect of Alcohol Drinking"

_healthcare, 2023, doi:10.3390/healthcare11202711_

Round 1

Reviewer 1 Report

This is an interesting research about the association between suicide ideation, depression and alcohol consumption. I have some comments aimed at improving the manuscript.

In lines 21 and 22 the authors mention an increase in suicides in South Korea, but they fail to include the death rate because with absolute measures we do not know if mortality increased or not. I suggest they include the death rate.

The manuscript has many orthographic errors. This should be addressed. I suggest a thorough English revision.

I suggest that the title includes that the study was carried out in South Korea.

The statement “As such, patients with chronic mental disorders have far more 40 suicidal ideation than the general population” (line 40 to 41) does not have a literature reference.

The authors search to analyze the mediating effect of alcohol drinking on the relationship between depression and suicide ideation. But the authors also state that alcohol drinking also affects depression (lines 58 and 59). How is this relationship addressed in this study as alcohol is only considered as a mediating variable between depression and suicide? I suggest this is clearly addressed. The methods used to analyze the mediating effect of alcohol are limited, I suggest that a Structual-Equation Model is used to address this relationship and the possible reverse association between depression and alcohol is correctly analyzed.

The authors mention that among the individuals that volunteered to participate, the subjects who were judged to be able to participate in the survey were chosen. But it is never stated which is the criteria to choose whether an individual should participate or not. Which is the inclusion/exclusion criteria?

The tables are not presented in order.

In section 3.4 the Step 3 explanations are different (lines 212 and 222). This should be clarified.

The authors propose some limitations of their analysis, but fail to address that as it is a cross-section study the order of the variables is not possible to be known. I mean that we don’t know if alcohol consumption predates depression or if suicide ideation came before alcohol consumption, and so on. The authors can only address associations but not causation which in itself is a limitation of the analysis. As I stated before, the effect of alcohol consumption on depression is also not analyzed in this study. Also it is not clear if the study subjects have other psychiatric disorders (other than depression and anxiety) because the inclusion criteria are not clearly stated. This is also a limitation because people with more than one psychiatric disorder can have a higher risk of suicide ideation, this should be at least acknowledged.

The manuscript has many orthographic errors. This should be addressed. I suggest a thorough English revision.

Author Response

I checked and revised the thesis as a whole. The modified part is marked in red on the attachment. I have responded to the critical points in the review in the following way. The comments are italicized.

Reviewer 2 Report

I read the proposed paper with great interest. Suicide is one of the major public health problems in the world, and the proposed study helps to implement knowledge of it for future preventive strategies.

Some minor comments:

- There are numerous typos (lack of spaces between words, probably a problem with the pdf file). The easy readability of the entire manuscript is compromised.

- The authors should better clarify why they excluded specific individuals. How did they figure out who did not answer the questionnaire honestly?

- There are formatting problems with Table 3.

- The entire study was conducted during the COVID pandemic that is known to have impacted people's mental health. This would merit further discussion in the discussions.

- The authors should include the limitations of the study

There are numerous typos (lack of spaces between words, probably a problem with the pdf file). The easy readability of the entire manuscript is compromised

Author Response

I checked and revised the thesis as a whole. The modified part is marked in red on the attachment. I have responded to the critical points in the review in the following way. The comments are italicized

Reviewer 3 Report

The study has potential but the format is poor. The abstract includes elements such as the statistical packet used which is not esential and should be included in the text. In the abstract, the authors conclude that to decrease levels of suicidal ideation in psychiatric patients, it is necessary to use interventions decreasing levels of alcohol drinking. I suggest a phrasing around the benefits of considering this type of intervention.

In the methods section, basic elements such as the inclusion/exclusion criteria of participants are not mentioned.

The introduction and the discussion need to be improved

It can be improved

Author Response

(The authors gave the same response as above.)

Reviewer 4 Report

This is a very interesting paper studying the relationship between depression, suicide and alcohol. The analysis is interesting, even though the text needs some improvement when citing the results. 

In this section, all the paragraphs are completely technical, without a special focus on the results or the story's narration. Maybe some improvements may take place here. 

Another question regards the psychological situation of the subjects. They asked themselves about their psychological condition. I wonder if the authors had direct access to medical data for each subject. Also, the authors do not discuss at all the degree of accuracy of the responses. 

A final question is related to the characteristics of the subject. All forms of depression have an equal probability of suicide?

Nevertheless, it is a very interesting paper, and I suggest that it must be published, after authors make some corrections. 

The authors must check the whole document. The spaces between different words sometimes vanish. I also think that during this process they will improve the overall quality of their expression.

Author Response

(The authors gave the same response as above.)

Round 2

Reviewer 1 Report

While the authors have addressed most of the comments, a couple of suggestions still require correction.

The authors aimed to analyze the mediating effect of alcohol drinking on the relationship between depression and suicide ideation. However, the authors previously stated in the previous version that alcohol drinking also affects depression, which influenced their proposed model. In this revised version, the authors simply removed the reference instead of addressing the problem. I suggest that at least this reverse association between depression and alcohol be mentioned as a limitation of the study.

Furthermore, I mentioned in my previous review that the authors "fail to address that as it is a cross-section study the order of the variables is not possible to be known. I mean that we don’t know if alcohol consumption predates depression or if suicide ideation came before alcohol consumption, and so on. The authors can only address associations but not causation which in itself is a limitation of the analysis." The response was that future studies should aim to confirm causal relationships through longitudinal studies. I agree that this is the correct way to state that limitation, but it is never explicitly mentioned in the manuscript. Therefore, I suggest that it be included explicitly as another limitation and as a suggestion for future research.

The paragraph included in section 2.2 contains numerous orthographic mistakes that need to be corrected.

Author Response

(The authors gave the same response as above.)
